# Effects of One-Year Simulated Nitrogen and Acid Deposition on Soil Respiration in a Subtropical Plantation in China

**Shengsheng Xiao [1,2,\*], G. Geoff Wang [3], Chongjun Tang [1,2], Huanying Fang [2], Jian Duan [1,2] and Xiaofang Yu [4]**

[1]   Jiangxi Provincial Key Laboratory of Soil Erosion and Prevention, Nanchang 330029, China; tangchongjun@126.com (C.T.); duanjian8807@163.com (J.D.)
[2]   Jiangxi Institute of Soil and Water Conservation, Nanchang 330029, China; fanghuanying@126.com
[3]   Department of Forestry and Natural Resources, Clemson University, Clemson, SC 29634-0317, USA; gwang@g.clemson.edu
[4]   School of Geography and Environment, Jiangxi Normal University, Nanchang 330029, China; sunnymissyu@gmail.com
\*   Correspondence: xss19811213@163.com

**Abstract:** Atmospheric nitrogen (N) and acid deposition have become global environmental issues and are likely to alter soil respiration ($R_s$); the largest $CO_2$ source is from soil to the atmosphere. However, to date, much less attention has been focused on the interactive effects and underlying mechanisms of N and acid deposition on $R_s$, especially for ecosystems that are simultaneously subjected to elevated levels of deposition of both N and acid. Here, to examine the effects of N addition, acid addition, and their interactions with $R_s$, we conducted a two-way factorial N addition (control, CK; 60 kg N ha$^{-1}$ a$^{-1}$, LN; 120 kg N ha$^{-1}$ a$^{-1}$, HN) and acid addition (control, CK; pH 4.5, LA; pH 2.5, HA) field experiment in a subtropical plantation in China. Our results showed the following: (1) During the one-year observation period, the seasonal dynamics of $R_s$ presented a single peak curve model, which was closely related to the surface soil temperature. (2) The simulated N deposition and acid deposition significantly decreased the $R_s$ in the subtropical plantation. Compared to the CK plots, the LN and HN treatments reduced the annual mean values of $R_s$ by 41% and 56%, and the annual mean values of $R_s$ were inhibited by 26% and 31% in the LA and HA plots. The inhibition of N application on $R_s$ was stronger than that of the simulated acid deposition. (3) Significant interactions between N addition and acid addition on $R_s$ were detected, and $R_s$ was significantly inhibited under four co-addition treatments. (4) The underlying mechanism and main reason for the responses of $R_s$ to simulated N and acid deposition in this study might be the inhibition of soil microbial biomass and soil enzyme activity due to soil acidification under increased N and acid input.

**Keywords:** nitrogen deposition; acid deposition; soil respiration; subtropical forest

---

## 1. Introduction

Over the past century, atmospheric nitrogen (N) and acid deposition have become global environmental issues, and climate change models predict that atmospheric N deposition and acid deposition will continue to increase within subtropical regions [1–3]. In the coming 20 years, worldwide N deposition is anticipated to increase by between 50% and 100% by 2030 compared to that in 2000, with the largest absolute increases occurring over East and South Asia [2], especially in China [4]. In China, the fastest increase rate of N deposition has been documented in subtropical areas, such as central and southeast China [5]. The atmospheric wet/bulk N deposition in subtropical China

ranges from 26 to 55 kg N ha$^{-1}$ a$^{-1}$ [6,7]. Globally, acid rain deposition has recently declined in developed countries, but it is continuously increasing in developing regions and countries, particularly China, with acidic depositional areas accounting for 30% of its land area [3]. In the 2010s, the average precipitation pH value was 4.70 in China [3]. Consequently, the center of the most severe acid rain area, the area south of the Yangtze River, moved eastwards [1,3]. Jia and Gao [8] also reported that the red soil region of southern China is known to be heavily polluted with acid deposition. In addition, some researchers have suggested that sulfur (S) deposition in China started to decrease as early as 2006, while N deposition continuous increased. The acid deposition in China is gradually changing from sulfuric acid rain to sulfuric–nitric compound acid precipitation [9].

The increased N availability and soil acidification under the boom of N and acid deposition have greatly affected terrestrial C cycling [10,11], especially the soil respiration (*Rs*) process [12–14], which is the largest CO2 efflux from terrestrial ecosystems to the atmosphere [15,16].

The responses of $R_s$ to simulated atmospheric N deposition have been well documented but are inconsistent and controversial among different terrestrial ecosystems. These responses mainly include promotion [17–22], inhibition [23–26], and no significant effect [14,27–30] due to the differences in vegetation types, soil conditions (especially the initial soil N levels), and the amount and time of N application that is regulated [18,31,32]. The effects of simulated acid deposition on *Rs* mainly manifest in two ways. First, there is no significant effect under a low acid application level [12,14]; second, there is inhibition under a high acid application level [33–37] because of the soil buffer's action on acid addition [38,39].

The processes and ecological effects of atmospheric N deposition and acid deposition occur simultaneously and interactively; therefore, it is extremely difficult to separate them [14]. In general, N and Acid deposition strengthen the interactions between factors and complicate the mechanisms of soil respiration under global changes. However, to date, few studies have examined both independent and interactive effects and the underlying mechanisms for the responses of *Rs* to N deposition and acid addition (most studies have only focused on simulated N deposition on *Rs*) [14,40,41]. This is an important research question, especially for ecosystems often subjected to elevated levels of deposition of both N and acid.

However, the response studies of $R_s$ to simulated atmospheric N deposition, acid deposition, and co-occurrence of N and acid application have concentrated on temperate natural forests [14]. The results from these studies may not be applicable to subtropical and tropical forests, given that biogeochemical cycles vary across forest types and climatic regions. The future intensity of N and acid deposition in subtropical and tropical regions is higher than that in temperate regions [1,3,5,10]. However, little attention has been paid to the effects of *Rs* on N and acid deposition in these ecosystems. Firstly, base cation depletion might be more intense in subtropical and tropical regions than in temperate regions, as this type of depletion is favored by high temperature and rain conditions. Secondly, compared to the wide-spread N limitations to forest productivity in temperate regions, N is not a limiting factor for most tropical and subtropical regions, as high N deposition [4] and nutrient limitations to productivity in these regions are typically driven by phosphorus [42,43]). Thus, responses of ecosystem processes to high N deposition rates in tropical and subtropical regions may be expressed differently than those in temperate forests. As Mo et al. [11] indicated, the effects of N deposition on the soil C cycles of tropical and subtropical forests are significantly different between N-enriched and N-limited forests. In addition, compared to temperate regions, such as northern China, the soil's buffering capacity for acid enrichment is stronger in south China due to its acidic soil and the long-term adaptations of microbes. It is necessary to clarify whether the effects of different spatial patterns of atmospheric N and acid deposition on $R_s$ are consistent in different regions [14].

In this study, we conducted a field experiment to simulate atmospheric N and acid deposition in a typical subtropical plantation in Chin to explore their effects on $R_s$. The main objectives of this study were to (1) investigate how $R_s$ responds to N deposition and acid deposition; and (2) to explore how soil environmental and biotic factors control the responses of $R_s$ to N deposition and acid deposition. Our

results are expected to provide novel insights into how N deposition and acid deposition interact to regulate $R_s$ and to emphasize the importance of evaluating these two processes together in the future.

## 2. Materials and Methods

### 2.1. Site Description

The study was conducted at the Jiangxi Eco-Science Park of Soil and Water Conservation (115°42′–15°43′ E, 29°16′–29°17′ N), about 5 km southeast of De'an county in Jiangxi province (Figure 1). The study area altitude ranges from 30 m to 90 m, and the area belongs to a subtropical humid monsoon climate zone, with an average annual temperature of 16.7 °C and an annual precipitation of 1469 mm. This region has approximately 245 to 260 frost-free days. The red soil in this region was primarily produced by the weathering of quaternary sediments and is classified as chestnut soil in the Chinese classification or Calcic–orthic Aridisol in the Soil Taxonomy classification.

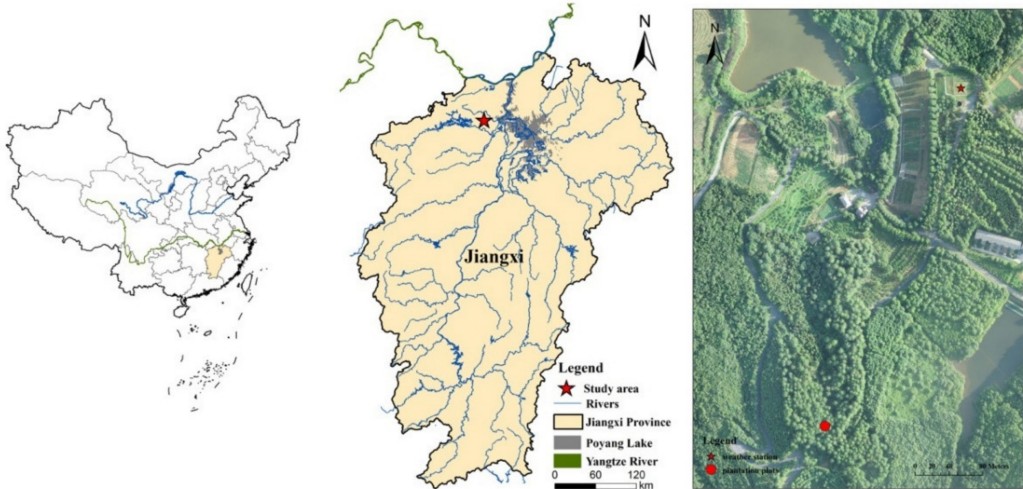

**Figure 1.** The study site location in the center of subtropical China. The study site was 15 km from Poyang Lake, the largest fresh water lake in China. The photo on the right shows the plantation plot and the automatic weather station.

In the park, about 50 ha of *Pinus elliottii* plantations were planted in the early 1980s for soil erosion control and ecological restoration. Their average diameter at breast height (*DBH*) values and tree heights were 20.4 cm and 15.7 m, respectively, with 85% coverage of understory vegetation, including *Schima superb*, *Adinandra millettii*, and *Dicranopteris dichotoma*. A previous study showed that the plantation plots had a soil erosion modulus of 0.12 t $km^{-2}$ $a^{-1}$, and the initial basic characteristics of the surface soils (0–20 cm) are as follows: soil pH: 6.27 ± 0.21; soil bulk density: 1.38 ± 0.03 g $cm^{-3}$; soil organic carbon: 15.52 ± 0.66 g $kg^{-1}$; soil total nitrogen: 1.03 ± 0.10 g $kg^{-1}$; $NH_4^+$: 19.42 ± 0.44 mg $kg^{-1}$; and $NO_3^-$: 26.42 ± 1.10 mg $kg^{-1}$ [44]. The *Pinus elliottii* plantations in this area remain relatively stable after 35 years of growth, providing suitable natural plots for this study. As reported by a previous monitor [44], this region has been subjected to serious atmospheric N and acid deposition, accompanying increasing emissions of $NO_x$, $NH_3$, and $SO_4^{2-}$ in Southern China. The natural N deposition intensity in the study area is about 43 kg N $ha^{-1}$ $a^{-1}$ (rainfall wet deposition), and the average pH value of the rainfall is 5.2.

### 2.2. Experimental Design

The experimental treatments included a control (CK) plot, a plot with N addition and 60 kg N $ha^{-1}$ $a^{-1}$ (LN)/120 kg N $ha^{-1}$ $a^{-1}$ (HN), a plot with acid addition and a pH of 4.5 (LA) and 2.5 (HA), and a plot featuring the addition of both N and Acid with $N_{60}$ + $A_{4.5}$ (LNLA), $N_{60}$ + $A_{2.5}$ (LNHA), $N_{120}$ + $A_{4.5}$ (HNLA), and $N_{120}$ + $A_{2.5}$ (HNHA). Each treatment had three replicates. Overall, 27 experimental

plots were established with a random block design of 7 m × 7 m (the inner 5 m × 5 m was the core area, and the outer 2 m × 2 m was the buffer area; the buffer area was treated same as the core area for research activities). It should be noted that the plots with relatively high concentration additions were established downhill to minimize the impact of N and acid migration caused by surface runoff. The N addition rates were chosen based on the reported atmospheric N deposition rate nearby and the forecasted N deposition rate in the future [6,7,45]; the same was true for acid deposition [1,3,8]. In order to clearly determine the future responses of soil respiration to the boom of atmospheric N deposition and acid rain aggravation in different stages, all experimental treatments were performed under the background of natural rainfall, and the N and acid added in the experiment were incremental.

Nitrogen was added via $NH_4NO_3$ fertilizer mixed with deionized water. Acid treatment was performed by mixing a concentration of 98% sulfuric acid ($H_2SO_4$) and a concentration of 68% nitric acid ($HNO_3$) with a 3:1 molar ratio, and then diluting the acid solution to a pH of 4.5 and 2.5 by mixing an appropriate amount deionized water. Currently, the acid rain in South China is mainly sulfuric acid rain, but there is a trend toward a gradual transformation to mixed acid rain containing sulfuric acid and nitric acid, with the gradual control of sulfur dioxide and the increase in the emissions of human-made active N. At present, the ratio of sulfates and nitrates in acid rain is generally 6:1–2:1. All the N and acid addition treatments were conducted the same as the rainfall pattern because the N deposition and acid deposition are mainly wet depositions in Southern China [46]. Long-term rainfall data for the automatic weather station in the park showed that the rainfall in April–July (rainy season) accounts for 50% of the annual rainfall, and 20% of precipitation occurs between August and October, mainly because of typhoons, while only 30% of precipitation occurs from November to March [47]. Therefore, the total N and acid application were both divided into three parts, which were applied on April 16th, August 24th, and November 16th in 2015, according to the proportions of 50%, 20%, and 30%. Each N or acid addition solution with 50 L deionized water (equivalent to rainfall of 1 mm) was sprayed uniformly on each plot with a spray pot. Similarly, the same amount of water was added to the CK plots.

*2.3. Soil Respiration Measurement*

Three polyvinyl chloride (PVC) collars (with a 20.4 cm inside diameter and 10 cm in height) were randomly installed for soil respiration rate ($R_s$) measurements in each plot. All the ground surface living plants in the collars were cut away one day before each measurement to avoid interference from $CO_2$ through plant respiration. All three PVC pipes in each plot were exposed to the ground 2–3 cm deep (the accurate numbers were recorded for calculating the total volume of the chamber), with a rubber hammer being used to fix the collar outside the soil.

From January to December of 2015, soil surface $CO_2$ effluxes for $R_s$ were measured twice per month using an LI-COR 8100A automated soil $CO_2$ flux system (LI-COR Inc., Lincoln, NE, USA). It should be noted that the soil respiration measurement frequency increased to every one or two days after N/acid addition. Each collar was measured for two minutes, with 2 replicates on sunny days from 09:00 to 11:30. The soil temperatures at a 5 cm depth ($T_5$) and the soil volumetric water content at the top 10 cm depth (*SWC*) adjacent to each collar were simultaneously detected by a digital thermocouple and Time-Domain Reflectometry (CS620, Campbell Scientific, Logan, UT, USA), respectively.

*2.4. Soil Sample Collection and Determination*

In 2015, while monitoring the soil respiration rate, soil samples were collected eight times in total (once before and after each N/acid addition). Five soil samples at 0–5 cm and 5–10 cm soil depths in each plot were collected and combined as a composite sample for different soil layers. After each sampling, the soil samples were placed in a cooler, transported back to the laboratory, and stored in a refrigerator at −4 °C. Fresh soil samples were sieved through a 2 mm mesh, with visible roots, organic debris, and rocks removed; then the samples were divided into two subsamples. One was air-dried to determine the soil pH value, total organic carbon content (TOC), total nitrogen content (TN), and the

activity of soil invertase and soil urease (involved in C, N cycling) according the methods in Table 1; the other was stored at −4 °C for the determination of the content of $NO_3^-$, $NH_4^+$, dissolved organic carbon (DOC), dissolved organic nitrogen (DON), microbial biomass carbon (MBC), and microbial biomass nitrogen (MBN), which occurred no more than two weeks after sampling (Table 1).

**Table 1.** Soil sample determination methods for the main indexes.

| Indexes | Determination Method | References |
|---|---|---|
| pH | Using a pH meter (Orion, Thermo Fisher Scientific Inc., Beverly, MA, USA) at a 1-to-5 ratio of soil-to-deionized water | - |
| TN, $NO_3^-$, $NH_4^+$ | Soil samples were extracted using a 2 mol $L^{-1}$ KCl solution first (at 1-to-5 ratio of soil-to-KCl solution) and then measured by a continuous flow analyzer (AA3 Bran + Luebbe, Germany) | - |
| TOC | potassium dichromate oxidation method | - |
| DOC | Water extraction first (at a 1-to-10 ratio of soil-to-deionized water), filtered through a membrane filter with 0.45 μm pores, followed by determination via aa TOC analyzer (Vario TOC Cube, Elementar, Germany) | [48] |
| DON | The difference between extractable N and available N ($NO_3^-$ + $NH_4^+$) | - |
| MBC, MBN | was determined by the chloroform fumigation-extraction method within a week after sampling. A 0.5 mol $L^{-1}$ $K_2SO_4$ solution was used to extract C and N from fumigated and non-fumigated samples at a 1:10 (*w/v*) ratio, followed by determination via a TOC analyzer (Vario TOC Cube, Elementar, Germany) | [48] |
| soil invertase activity | Sodium phenol colorimetry | [49] |
| soil urease activity | 3,5-dinitrosalicylic acid colorimetry | [49] |

*2.5. Statistical Analysis*

Before performing the ANOVA, the normality of distribution was checked by a Kolmogorov–Smirnov test and homogeneity of variance was determined by Levene's test. All data passed these tests and no transformation was needed. A multi-way analysis of variance (ANOVA) was used to determine the main effects and interactive effects of N addition and acid addition on the $R_s$ and soil variables. Differences in the $R_s$ and soil variables among the different experimental treatments were analyzed using a one-way analysis of variance (ANOVA) with Tukey's HSD test. Pearson's correlation analysis was performed to determine the relationship between $R_s$ and the soil variables, as well as among different soil variables. It should be pointed out that when measuring the correlations between $R_s$ and soil pH, soil enzymes activity and other soil C, N variables, all soil respiration rates data had to be divided into eight groups according to the soil sampling time and the simulated deposition time of N and acid; the average values for the $R_s$ of every group were used. All ANOVA and correlation analyses were conducted with a significance criterion of $p < 0.05$ (unless otherwise declared) using IBM SPSS 19 statistical software (SPSS, Chicago, IL, USA), and graphs were prepared with Origin 8.5 (Origin Lab Corporation, Northampton, USA).

## 3. Results

*3.1. N and Acid Addition Effects on $R_s$ Dynamics and Average Annual Efflux*

The soil respiration rates showed clear seasonal dynamics with similar single peak curves in all experimental treatments (Figure 2c). Across the year, $R_s$ generally tracked $T_5$ (Figure 2a). The rates of $R_s$ increased with an increase of $T_5$ in all plots and reached its maximum in July or August. However, from the beginning of April, the increased margins under N addition or acid addition were obviously lower than those of the CK plots (Figure 2c). Throughout the one-year period of study, the rates $R_s$ were inhibited by N or acid addition.

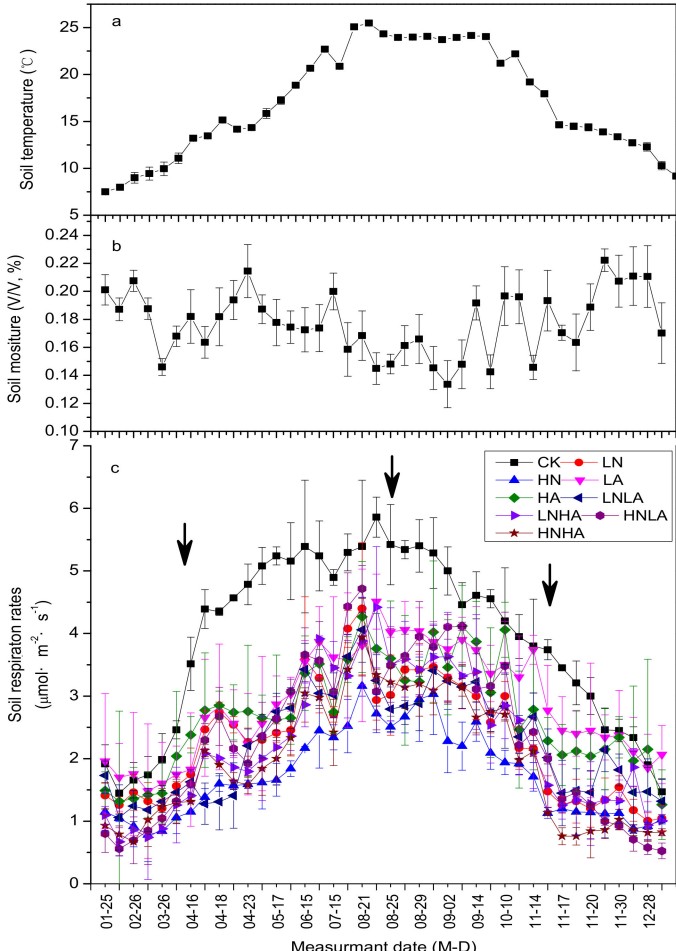

**Figure 2.** Monthly dynamics of soil temperature at a 5 cm soil depth (**a**), soil water content at a 10-cm soil depth (**b**), and the soil respiration rate (**c**) in 2015. Notes: Error bars indicate the standard error ($n = 9$ in photo "a" and "b", $n = 3$ in photo "c"). The three black arrows in photo C show the three additions of N and acid on April 16th, August 24th, and November 16th in 2015. CK: control; LN: low N addition; HN: high N addition; LA: low acid addition; HA: high acid addition; LNLA: both the addition of low N and low acid; LNHA: low N addition plus high acid addition; HNLA: high N addition plus low acid addition; HNHA: both the addition of high N and high acid (the same below). The data for soil temperature and moisture dynamic curves are the means under nine treatments because of the small difference between different treatments.

### 3.2. N and acid Addition Effects on Average Annual Efflux

A two-factor variance analysis showed that both N addition and acid addition had significant effects on the rates of $R_s$, and there were also significant interactive effects between N addition and acid addition (Table 2, $p = 0.000$). These effects can also be derived from Figure 3B. Across the year, the mean values of $R_s$ (mean ± SE, $n = 40$, µmol m$^{-2}$·s$^{-1}$) among the nine treatments demonstrated CK (3.91 ± 0.22) > LA (2.90 ± 0.13) > HA (2.70 ± 0.13) > HNLA (2.35 ± 0.20) > LN (2.30 ± 0.15) > LNLA (2.27 ± 0.13) > LNHA (2.24 ± 0.17) > HNHA (1.97 ± 0.16) > HN (1.73 ± 0.11). Compared to the CK plots, the annual mean $R_s$ was significantly inhibited under N addition. The $R_s$ decreased by 41% and 56% in the LN and the HN treatments, respectively. There were significant differences among the three treatments (Figure 3b). Compared to CK, the treatments of LA and HA significantly decreased $R_s$ by 25.94% and 30.93% (Figure 3b); however, there were no significant differences in $R_s$ between the LA and the HA treatments (Figure 3b). There were significant differences between LN and LA, HN and LA, HN and HA. Furthermore, compared to the CK treatment plots, the fluxes of $R_s$ were significantly inhibited under the combined effects of N and acid addition, but there was no significant difference

among the four kinds of compound additions. However, the $R_s$ values under the compound addition treatments were significantly higher than the $R_s$ of the HN treatment and the LA treatment, which indicated that there were obvious interactions between N addition and acid addition.

**Table 2.** Main effects and interactive effects of N addition and acid addition on soil respiration rate and soil properties via a two-factor ANOVA. N: N addition; A: acid additions; N*A: N addition plus acid addition; data are shown at significance levels. * and ** indicate significant differences at 0.05 ($p < 0.05$) and 0.01 significance levels ($p < 0.01$), respectively. $R_s$, TOC, DOC, DON, MBC and MBN Respectively refer to soil respiration rate, total organic carbon, dissolved organic carbon, dissolved organic nitrogen, microbial biomass carbon, and microbial biomass nitrogen.

| | $R_s$ | 0–5 cm | | | | | | | | | |
|---|---|---|---|---|---|---|---|---|---|---|---|
| | | pH | Soil Urease Activity | Soil Invertase Activity | TOC | $NH_4^+$ | $NO_3^-$ | DOC | DON | MBC | MBN |
| **N** | 0.000 ** | 0.003 ** | 0.004 ** | 0.209 | 0.373 | 0.898 | 0.055 | 0.999 | 0.260 | 0.849 | 0.756 |
| **A** | 0.031 * | 0.005 ** | 0.000 ** | 0.545 | 0.427 | 0.496 | 0.914 | 0.999 | 0.980 | 0.146 | 0.978 |
| **N*A** | 0.000 ** | 0.206 | 0.962 | 0.381 | 0.509 | 0.840 | 0.824 | 0.917 | 0.976 | 0.183 | 0.244 |
| | | 5–10 cm | | | | | | | | | |
| | | pH | Soil Urease Activity | Soil Invertase Activity | TOC | $NH_4^+$ | $NO_3^-$ | DOC | DON | MBC | MBN |
| **N** | | 0.022 * | 0.018 * | 0.170 | 0.448 | 0.504 | 0.067 | 0.986 | 0.833 | 0.322 | 0.858 |
| **A** | | 0.000 ** | 0.001 ** | 0.476 | 0.609 | 0.515 | 0.375 | 0.995 | 0.963 | 0.022 * | 0.151 |
| **N*A** | | 0.018 * | 0.793 | 0.342 | 0.103 | 1.000 | 0.362 | 0.951 | 0.967 | 0.335 | 0.506 |

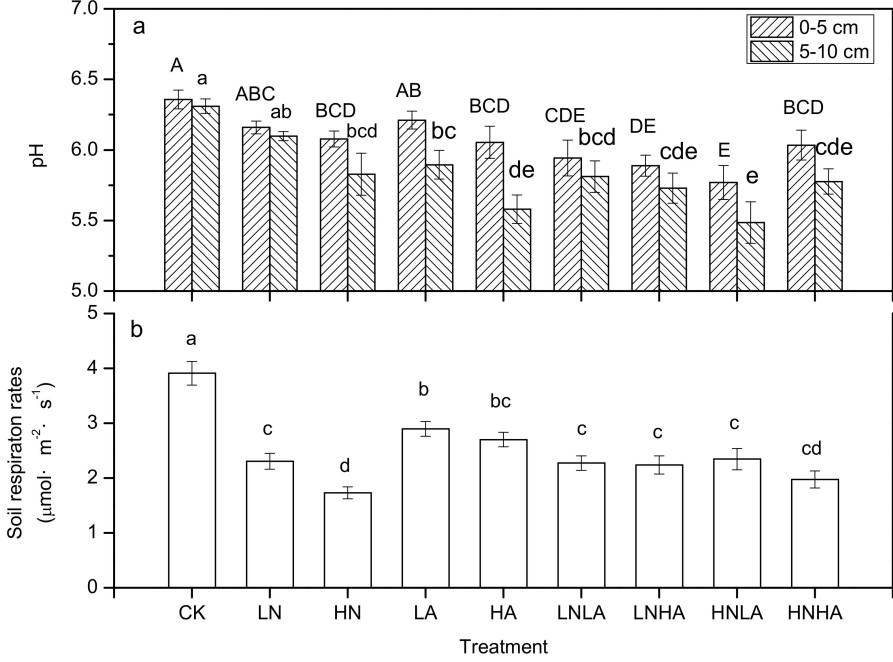

**Figure 3.** Effects of the experimental treatments on the soil pH (**a**) and annual soil respiration rate (**b**). Error bars indicate standard errors ($n = 8$ in photo "a" and $n = 40$ in photo "b"). Different lowercase letters under different treatments indicate significant differences at a $p < 0.05$ level.

### 3.3. N and Acid Addition Effects on Soil Enzyme Activity and Soil Properties

According to the two-factor ANOVA, N and acid addition had significant effects on the activity of soil urease in both the 0–5 cm and 5–10 cm layers (Table 2) but exerted no significant effect on soil invertase activity in the two soil layers (Table 2). Furthermore, the interaction between the N

addition and acid addition had no significant effect on the activity of soil urease or invertase (Table 2). In general, soil urease was more sensitive to the addition of N and acid than soil invertase. Soil urease activity decreased gradually with an increase in N and acid addition levels, and compound addition treatments also significantly inhibited the activity of soil urease in both soil layers (Figure 4a). For soil invertase activity, only significant differences between HN and CK, LA and CK, and HNHA and CK were detected in both soil layers (Figure 4b). In addition, the activities of soil urease and invertase in the 0–5 cm layer were higher than those in the 5–10 cm layer (Figure 4).

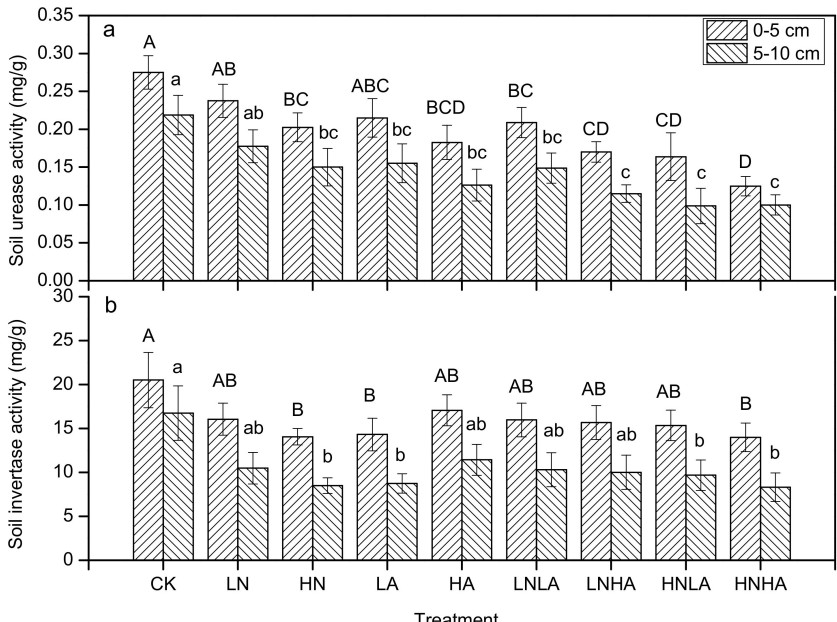

**Figure 4.** Effects of N and acid addition on the activity of soil urease (**a**) and invertase (**b**). Error bars indicate a standard error (*n* = 8). Different lowercase letters indicate significant differences at a *p* < 0.05 level under different treatments.

Nitrogen and acid addition had significant effects on soil pH in both the 0–5 cm and 5–10 cm soil layers (Table 2). Furthermore, the interaction of N and acid addition had no significant effect on the soil pH at 0–5 cm but had a significant effect at 5–10 cm (Table 2). Soil pH fluctuated by 5.15–6.72 in the 0–5 cm layer and by 4.91–6.49 in the 5–10 cm soil layer for all treatments (Figure 3a). The soil pH of the lower layer was lower than that of the upper soil layer (Figure 3a). Compared to the CK plots, HN treatments significantly reduced the soil pH in the two soil layers (Figure 3a). Significant differences were detected between the CK and LA treatments only at 0–5 cm, but between the CK and HA treatments in the two soil layers, the co-addition of N and acid significantly reduced the soil pH (Figure 3a). These results show that the red soil of our plots had weak acidity, and the addition of N and acid aggravated the soil acidity, especially under the HN treatment, HA treatment, and the compound additions of N and acid.

In general, the responses of different C and N indicators to N or acid addition were different. For TOC, there were significant differences between HN and CK, HA and CK, and HNHA and CK in the 5–10 cm soil layer (Table 3). Soil $NH_4^+$ content was not significantly altered by either N addition or acid addition treatments in the two soil layers, but HA treatments significantly reduced soil $NO_3^-$ content in the 5–10 cm layer (Table 3). Soil DOC and DON contents were also not significantly altered by either N addition or acid addition treatments (Table 3). Four additional co-treatments significantly reduced the soil MBC content in the two soil layers (Table 3). For soil MBN content, there was no significant difference among all treatments in the 0–5 cm layer, but significant differences were observed between HA and CK and LNLA and CK in the 5–10 cm layer (Table 3).

**Table 3.** Soil properties in the top two soil layers under different experimental treatments. Data are the means ± standard error ($n = 8$); different letters in one row indicate a significant difference under different treatments at a 0.05 level ($p < 0.05$).

| Soil Layer | Parameter | Treatment | | | | | | | | |
|---|---|---|---|---|---|---|---|---|---|---|
| | | CK | LN | HN | LA | HA | LNLA | LNHA | HNLA | HNHA |
| 0~5 cm | TOC (g kg$^{-1}$) | 32.1 ± 5.0 [a] | 30.8 ± 3.4 [a] | 33.5 ± 3.0 [a] | 30.9 ± 1.6 [a] | 27.3 ± 3.1 [a] | 33.0 ± 3.9 [a] | 32.5 ± 4.3 [a] | 25.0 ± 1.2 [a] | 26.2 ± 2.9 [a] |
| | NH$_4^+$ (mg kg$^{-1}$) | 4.86 ± 0.66 [a] | 5.96 ± 2.02 [a] | 5.98 ± 0.87 [a] | 5.15 ± 1.58 [a] | 6.87 ± 1.47 [a] | 4.87 ± 0.61 [a] | 7.69 ± 2.66 [a] | 5.79 ± 1.24 [a] | 5.48 ± 1.32 [a] |
| | NO$_3^-$ (mg kg$^{-1}$) | 17.39 ± 5.99 [a] | 28.42 ± 6.99 [a] | 34.10 ± 5.75 [a] | 21.23 ± 4.19 [a] | 19.39 ± 4.82 [a] | 28.18 ± 9.91 [a] | 31.73 ± 6.26 [a] | 25.18 ± 5.11 [a] | 28.51 ± 5.55 [a] |
| | DOC (mg kg$^{-1}$) | 174.01 ± 51.90 [a] | 196.29 ± 48.89 [a] | 225.33 ± 53.96 [a] | 215.26 ± 53.14 [a] | 209.29 ± 59.13 [a] | 190.63 ± 48.99 [a] | 214.02 ± 55.99 [a] | 194.08 ± 50.69 [a] | 175.50 ± 44.18 [a] |
| | DON (mg kg$^{-1}$) | 14.92 ± 3.23 [a] | 25.17 ± 8.24 [a] | 21.75 ± 6.90 [a] | 17.12 ± 3.42 [a] | 17.79 ± 3.71 [a] | 25.37 ± 9.78 [a] | 24.43 ± 8.54 [a] | 16.17 ± 3.80 [a] | 18.14 ± 6.28 [a] |
| | MBC (mg kg$^{-1}$) | 165.62 ± 29.21 [ab] | 153.57 ± 36.96 [ab] | 230.01 ± 33.81 [a] | 172.93 ± 29.22 [ab] | 147.54 ± 28.28 [ab] | 129.16 ± 29.27 [b] | 160.79 ± 32.44 [ab] | 115.13 ± 20.29 [b] | 114.65 ± 32.81 [b] |
| | MBN (mg kg$^{-1}$) | 52.99 ± 6.14 [a] | 64.11 ± 18.39 [a] | 80.75 ± 10.29 [a] | 75.65 ± 11.61 [a] | 76.49 ± 8.79 [a] | 58.76 ± 9.83 [a] | 62.67 ± 7.57 [a] | 58.96 ± 5.95 [a] | 59.54 ± 14.07 [a] |
| 5~10 cm | TOC (g kg$^{-1}$) | 22.6 ± 3.2 [a] | 17.5 ± 1.8 [ab] | 15.7 ± 1.9 [b] | 16.8 ± 1.7 [ab] | 15.9 ± 2.7 [b] | 17.9 ± 1.4 [ab] | 19.9 ± 2.6 [ab] | 19.0 ± 1.3 [ab] | 14.9 ± 1.1 [b] |
| | NH$_4^+$ (mg kg$^{-1}$) | 5.14 ± 1.09 [a] | 7.45 ± 1.79 [a] | 6.73 ± 2.47 [a] | 6.32 ± 1.98 [a] | 7.42 ± 1.89 [a] | 8.16 ± 1.79 [a] | 9.71 ± 3.41 [a] | 7.91 ± 2.37 [a] | 8.69 ± 3.00 [a] |
| | NO$_3^-$ (mg kg$^{-1}$) | 11.80 ± 2.04 [a] | 13.97 ± 4.15 [a] | 11.41 ± 4.26 [a] | 6.83 ± 2.41 [ab] | 1.43 ± 0.49 [b] | 9.91 ± 3.62 [ab] | 15.23 ± 5.19 [a] | 11.51 ± 2.96 [a] | 9.76 ± 2.35 [ab] |
| | DOC (mg kg$^{-1}$) | 183.51 ± 50.05 [a] | 152.49 ± 40.35 [a] | 157.21 ± 38.12 [a] | 164.21 ± 37.58 [a] | 150.70 ± 39.29 [a] | 152.81 ± 35.20 [a] | 176.57 ± 42.97 [a] | 175.09 ± 40.63 [a] | 157.25 ± 35.39 [a] |
| | DON (mg kg$^{-1}$) | 13.79 ± 3.45 [a] | 13.24 ± 3.16 [a] | 13.11 ± 2.63 [a] | 12.36 ± 2.23 [a] | 11.04 ± 2.34 [a] | 12.34 ± 2.44 [a] | 13.13 ± 3.34 a | 13.93 ± 2.60 [a] | 14.25 ± 2.77 [a] |
| | MBC (mg kg$^{-1}$) | 140.12 ± 26.95 [a] | 94.64 ± 23.72 [ab] | 98.16 ± 18.11 [ab] | 64.47 ± 10.19 [b] | 97.57 ± 19.53 [ab] | 54.43 ± 16.22 [b] | 90.46 ± 17.10 [ab] | 84.03 ± 21.94 [b] | 60.84 ± 9.10 [b] |
| | MBN (mg kg$^{-1}$) | 54.17 ± 9.11 [a] | 43.04 ± 9.42 [ab] | 39.99 ± 6.84 [ab] | 37.89 ± 7.88 [ab] | 29.86 ± 3.06 [b] | 30.70 ± 5.74 [b] | 38.20 ± 7.86 [ab] | 40.20 ± 6.69 [ab] | 36.31 ± 7.70 [ab] |

### 3.4. Relationships Between $R_s$ and Soil Properties

There was a significant linear positive correlation between $R_s$ and $T_5$ ($r = 0.778$, $p < 0.001$) and a significant linear negative correlation between $R_s$ and $SWC$ ($r = -0.206$, $p < 0.001$) across the whole experimental period (Table 4). Soil respiration rates increased with an increase in pH at 0–5 cm and decreased with an increase in pH at 5–10 cm to some extent, but not significantly. Soil respiration rates had significantly positive correlations with the activity of soil urease and soil invertase in both soil layers, mostly at a significant level of 0.001. The rates of $R_s$ were linear negatively correlated to the content of TOC, $NH_4^+$, $NO_3^-$, DOC, DON, MBC and MBN. Especially in the 0–5 cm soil layer, the negative correlation reached a significant level ($p < 0.05$ or $p < 0.001$, Table 4).

**Table 4.** Pearson's correlation between soil respiration, climate factors, and soil properties. $R_S$: soil respiration, T: soil temperature at a 10 cm soil depth, SWC: soil water content at a 10 cm soil depth, MB: microbial biomass, * $p < 0.05$, ** $p < 0.01$.

| | | $R_s$ | $T_5$ | SWC | 0–5 cm pH | Soil Urease Activity | Soil Invertase Activity | TOC | $NH_4^+$ | $NO_3^-$ | DOC | DON | MBC | MBN | 5–10 cm pH | Soil Urease Activity | Soil Invertase Activity | TOC | $NH_4^+$ | $NO_3^-$ | DOC | DON | MBC | MBN |
|---|---|---|---|---|---|---|---|---|---|---|---|---|---|---|---|---|---|---|---|---|---|---|---|---|
| | $R_s$ | 1.000 | 0.778 ** | −0.206 ** | 0.157 | 0.746 ** | 0.800 * | −0.302 * | −0.393 ** | −0.590 * | −0.430 * | −0.289 * | −0.209 | −0.141 | −0.205 | 0.726 ** | 0.793 ** | −0.080 | −0.463 ** | −0.226 | −0.330 ** | −0.199 | −0.058 | 0.025 |
| | $T_5$ | | 1.000 | −0.247 ** | −0.031 | 0.657 ** | 0.777 ** | −0.356 ** | −0.487 ** | −0.545 ** | −0.612 ** | −0.330 ** | −0.312 ** | −0.081 | −0.321 ** | 0.671 ** | 0.752 ** | −0.188 | −0.526 ** | −0.314 ** | −0.597 ** | −0.402** | −0.208 | −0.110 |
| | SWC | | | 1.000 | −0.243 * | −0.501 ** | −0.294 * | −0.072 | 0.172 | 0.110 | 0.115 | −0.013 | −0.003 | −0.204 | −0.095 | −0.488 ** | −0.282 * | 0.158 | 0.138 | 0.129 | 0.202 | 0.196 | 0.164 | 0.147 |
| 0–5 cm | pH | | | | 1.000 | 0.174 | 0.007 | 0.306 ** | −0.010 | −0.002 | 0.058 | 0.176 | 0.260* | 0.289 * | 0.562 ** | 0.179 | −0.003 | 0.176 | −0.206 | 0.125 | −0.001 | 0.089 | 0.278 * | 0.219 |
| | soil urease activity | | | | | 1.000 | 0.713 ** | −0.205 | −0.308 ** | −0.490 ** | −0.353 ** | −0.203 | −0.178 | −0.161 | 0.017 | 0.939 ** | 0.694 ** | −0.125 | −0.277 * | −0.246 * | −0.306 ** | −0.249 * | −0.057 | −0.120 |
| | soil invertase activity | | | | | | 1.000 | −0.321 ** | −0.486 ** | −0.491 ** | −0.412 ** | −0.223 | −0.351 ** | −0.071 | −0.174 | 0.744 ** | 0.986 ** | −0.177 | −0.326 ** | −0.256* | −0.367 ** | −0.294 * | −0.172 | −0.030 |
| | TOC | | | | | | | 1.000 | 0.139 | 0.488 ** | 0.191 | 0.295 * | 0.475 ** | 0.313 ** | 0.326 ** | −0.233 * | −0.331 ** | 0.56 ** | −0.121 | 0.314 ** | 0.071 | 0.124 | 0.242 * | 0.221 |
| | $NH_4^+$ | | | | | | | | 1.000 | 0.335 ** | 0.442 ** | 0.302 ** | 0.241 * | 0.002 | 0.139 | −0.351 ** | −0.450 ** | 0.130 | 0.529 ** | 0.377 ** | 0.484 ** | 0.371 ** | 0.310 ** | 0.072 |
| | $NO_3^-$ | | | | | | | | | 1.000 | 0.495 ** | 0.481 ** | 0.402 ** | 0.372 ** | 0.141 | −0.498 ** | −0.490 ** | 0.301 * | 0.368 ** | 0.527 ** | 0.382 ** | 0.438 ** | 0.213 * | 0.219 |
| | DOC | | | | | | | | | | 1.000 | 0.581 ** | 0.512 ** | 0.128 | 0.227 | −0.319 ** | −0.379 ** | 0.005 | 0.537 ** | 0.342 ** | 0.901 ** | 0.763 ** | 0.431 ** | 0.165 |
| | DON | | | | | | | | | | | 1.000 | 0.346 ** | 0.218 | 0.310 ** | −0.166 | −0.213 | 0.151 | 0.257 * | 0.522 ** | 0.483 ** | 0.549 ** | 0.295 * | 0.199 |
| | MBC | | | | | | | | | | | | 1.000 | 0.272 * | 0.334 ** | −0.198 | −0.343 ** | 0.116 | 0.089 | 0.162 | 0.370 ** | 0.378 ** | 0.489 ** | 0.212 |
| | MBN | | | | | | | | | | | | | 1.000 | 0.131 | −0.171 | −0.084 | 0.233 * | −0.071 | 0.256 * | −0.011 | 0.109 | 0.172 | 0.372 ** |
| 5–10 cm | pH | | | | | | | | | | | | | | 1.000 | 0.033 | −0.156 | 0.219 | −0.100 | 0.406 ** | 0.219 | 0.289 * | 0.398 ** | 0.383 ** |
| | soil urease activity | | | | | | | | | | | | | | | 1.000 | 0.721 ** | −0.168 | −0.271 * | −0.258 * | −0.289 * | −0.222 | −0.086 | −0.128 |
| | soil invertase activity | | | | | | | | | | | | | | | | 1.000 | −0.168 | −0.308 ** | −0.239 * | −0.334 ** | −0.293 ** | −0.161 | −0.021 |
| | TOC | | | | | | | | | | | | | | | | | 1.000 | −0.202 | 0.436 ** | 0.057 | 0.114 | 0.365 ** | 0.556 ** |
| | $NH_4^+$ | | | | | | | | | | | | | | | | | | 1.000 | −0.027 | 0.595 ** | 0.332 ** | 0.012 | −0.219 |
| | $NO_3^-$ | | | | | | | | | | | | | | | | | | | 1.000 | 0.382 ** | 0.486 ** | 0.408 ** | 0.542 ** |
| | DOC | | | | | | | | | | | | | | | | | | | | 1.000 | 0.825 ** | 0.497 ** | 0.224 |
| | DON | | | | | | | | | | | | | | | | | | | | | 1.000 | 0.521 ** | 0.271 * |
| | MBC | | | | | | | | | | | | | | | | | | | | | | 1.000 | 0.482 ** |
| | MBN | | | | | | | | | | | | | | | | | | | | | | | 1.000 |

## 4. Discussion

### 4.1. Soil Respiration Response to N Addition

Because of the different initial soil N levels, vegetation types, soil microbial communities, and other parameters, in general, the effects of N deposition on $R_s$ are inconsistent and controversial among the different terrestrial ecosystems. A meta-analysis showed that the effects of simulated N deposition on $R_s$ in forest ranges from an inhibition of 57% to a promotion of 63% [25]. Throughout the one-year period monitored in this study, the rates of $R_s$ were significantly inhibited by N addition. This result is consistent with the results of Mo et al. [11], who studied a tropical broad-leaved forest, and Fan et al. [50], who obtained their results in a subtropical Chinese fir plantation.

Nitrogen enrichment induced changes in $R_s$ rates are jointly controlled by changes in an array of factors (both abiotic and biotic), especially changes in the microclimate, plant community composition and aboveground productivity, fine root biomass and productivity, and soil microbial biomass [51]. Overall, N deposition had two effects on the forest's ecosystem: a nutrient effect [19] and an acidification effect [23,36], which are contradictory. The ultimate effect of N addition on $R_s$ is determined by which is more advantageous: nutrition or acidification. Higher N availability under N deposition leads greater C allocated to aboveground tissues and less C allocated to fine roots [25], which will make the microbial carbon supply inadequate and inhibit microbial biomass [52] and microbial activity [53], thereby changing the soil's microbial community [36] and decreasing soil microbial respiration. This might be one reason for the responses of $R_s$ to the simulated N deposition.

Nitrogen addition had no significant effects on the content of TOC, $NH_4^+$, $NO_3^-$, DOC, and DON in the surface soil in this study (Table 3), which indicated that N input did not significantly change the respiratory substrate of microorganisms but might directly affect soil microorganisms themselves, which is supported by the conclusion that N treatments decreased soil MBC and MBN content to a certain extent (Table 3). Moreover, simulated N deposition decreased soil pH (Figure 3a) and had significant effects on the soil enzymes involved in soil organic matter transformation [54] (Figure 4). In addition, N input also inhibited root growth and reduced fine root biomass [54,55], thereby affecting root autotrophic respiration [34].

### 4.2. Effect of Acid Treatment on Soil Respiration

The effects of $R_s$ on the simulated acid deposition were relatively clear, with no significant effect at low acid application levels [12,14] and inhibition under high acid application levels [33–36] because of the soil buffer's action on acid [38,39]. In line with these observations [15,35,37,39], our one-year study also showed acid treatment to have significant inhibition effects on $R_s$ rates, with a lower pH having a more severe inhibitory effect. In another study in Dinghushan, Southern China, simulated acid rain inhibited $R_s$ in broadleaf and mixed coniferous and broad-leaved forests, with a lower pH having a more severe inhibitory effect [12]).

The influence of acid treatment on $R_s$ is not obvious, mainly due to the complex effects of soil buffering capacity in the short term [38]. However, strong or sustained acid deposition alters soil pH, ion exchange capacity, and chemical properties, thereby strengthening soil acidification, which, in turn, influence the activities of plant roots and soil microbes [56]. In this study, acid addition had significant effects on soil pH. Acid treatment caused the soil pH to decrease to 6.21 and 6.05, compared with the 6.36 of CK in the 0–5 cm soil layer, and to decrease to 5.90 and 5.58, compared with the 6.31 of CK in the 5–10 cm soil layer. Soil acidification is caused not only by direct acid rain input but also by the leaching process in red soil. A large amount of $H^+$ in precipitation falls into the soil, disrupting the soil's ion balance and leading to the run off containing more active $K^+$, $Ca^{2+}$, $Na^+$, $Mg^{2+}$, and other base ions. When precipitation exceeds evaporation, these soil base ions might be lost through the leaching processes, resulting in soil acidification [57], especially in the red soil region in Southern China due to the developed leaching phenomena [47].

In this study, accompanied by soil acidification, soil enzyme activity was inhibited, especially the activity of soil urease (Figure 4). Baath et al. [58] and Kim et al. [59] detected similar phenomena. Due to soil acidification, the long-term stress of alterations in the soil's microbial community composition and metabolic functions might affect the transformation and balance of soil C and N, thereby affecting the release of $CO_2$. Each microorganism has an optimum soil pH value range [15]. Microbial activity might be altered by soil H+ content [50]. Similarly, Baath et al. [58] also detected a significant decrease in metabolically active bacteria with high levels of acid addition, as well as a reduction in $CO_2$ release flux [36,60–62]. Wang et al. [61], Su et al. [36], and Kotas et al. [60] also drew similar conclusions and reported that soil acidification accompanied by increased acid deposition might inhibit $R_s$ by altering the microbial community structure and reducing microbial activity. In this study, only MBC and MBN were monitored. In future work, the responses of the microbial community structure to N addition also need to be monitored.

However, from another perspective, the limitations in the activity of soil microbes and enzymes may reduce the utilization rate of carbon sources by microorganisms, allowing a pool of relatively labile organic matter to accumulate, thus promoting the storage of soil organic C [35]. It can also be inferred that, with an increase in acid rain in the subtropical region of China, the decrease in forest $R_s$ rates is beneficial to the accumulation of soil organic C, which may be one of the mechanisms underlying old-growth forests accumulating carbon in their soils [22]. In addition, according to Liang et al. [34], fine root biomass was significantly correlated with $R_s$, and serious acid deposition could inhibit seedling growth, reduce fine root biomass, and inhibit root respiration. Thus, the direct evidence for fine root biomass measurements needs to be supplemented in future work.

### 4.3. Interaction Between N Addition and Acid Addition on Soil Respiration

In reality, N deposition and acid deposition are not completely independent but have strong interactions with each other. A certain amount of N input is accompanied by the precipitation of acid (acid rain), such as $NO_3^-$. At present, most acid rains have a $SO_4^{2-}/NO_3^-$ ratio of about 6:1–2:1 but tend to gradually change from a sulphuric acid dominant type to a sulfuric acid–nitric acid mixed type in China [1,3], which, in turn, will further increase the proportion of N input to the soil. On the other hand, N deposition also can potentially exacerbate soil acidification. Because of the elevated availability and mobility of $NO_3^-$, many base cations after a series of cation exchanges in soils will be lost, accompanied by $NO_3^-$ input to the soil [56,63].

Soil respiration is a complex process jointly regulated by both environmental and biological factors, including the climate, microbes, soil properties, and their interactions. The change direction and change degree of the $R_s$ rate can reflect the sensitivity and response mode of the plant–soil complex to environmental stresses, such as N deposition and acid deposition. However, although the independent effects of N deposition and acid addition on $R_s$ have been studied across a variety of forest types, but few studies have examined both the independent and interactive effects and underlying mechanisms of N deposition and acid addition on $R_s$.

In this study, compared to the CK plots, both N addition alone and acid addition alone decreased $R_s$ significantly. $R_s$ was significantly inhibited under the co-addition treatments of N and acid. The Pearson's correlation showed that there was no significant correlation between $R_s$ and the soil properties of C and N, but significant positive correlations between $R_s$ and the activity of soil urease and soil invertase were detected. Therefore, the inhibition of N addition plus acid addition on $R_s$ was not due to the limitation of the substrate supply to soil microbe respiration; instead, the main reason might be a change of the microorganisms themselves. The monitor data also showed that the MBC content in the two soil layers significantly decreased under the co-addition treatments (Table 3). Moreover, our research showed that the inhibitory effect of simulated N deposition on $R_s$ was greater than that of acid deposition (Table 2), which is inconsistent with the results of Zheng et al. [14], who found that, due to the low amount of N and acid addition, the inhibition effect under the combined addition of N and acid on $R_s$ did not reach a significant level.

In addition, simulated N and acid deposition had significant interactions and offsetting effects on $R_s$ in the subtropical plantation of our study. Liu et al. [64] found that soil sucrase activity was activated by severe acid rain, which might slow down the inhibitory effects of increasing N on soil respiration. However, in this study, the activity of soil sucrase was also inhibited, albeit with less sensitivity to N deposition than that of soil urease activity. As Wang et al. [65] reported, further long-term experiments with multiple levels of N and acid addition are needed to test the responses and underlying mechanisms of soil respiration to aggravating N and acid deposition.

## 5. Conclusions

Simulated N deposition, acid deposition, and the co-addition of N and acid significantly decreased the $R_s$ rates in a subtropical plantation in our study, which differed from temperate forests. In general, the addition of N and acid had no significant impact on soil C and N content but significantly reduced soil pH and decreased soil microbial biomass and soil enzyme activity to a certain extent, especially the activity of soil urease, which might be the main reason for the $R_s$ responses in this experiment.

Because the deposition processes and ecological effects of atmospheric N deposition and acid deposition occur simultaneously and interact with each other, their combined effects (rather than their individual effects) require more research emphasis. The findings in this research provide evidence of how the $R_s$ of a subtropical forest responds to N addition, acid addition, and their interactions, showing that the inhibitory effect of N application on $R_s$ was stronger than that of acid additions on $R_s$; simulating acid deposition enhanced their interaction effects. In the future, it will be important to distinguish the components of $R_s$ and extend the observation time, in order to further clarify the responses and internal mechanisms of forest $R_s$ under the background of serious atmospheric N and acid deposition, especially the response direction and response degree of root autotrophic respiration and microbial heterotrophic respiration to N and acid deposition, and their coupling.

**Author Contributions:** S.X. was responsible for funding acquisition and resources. G.G.W. and C.T. conceptualized the research. S.X., H.F., J.D., and X.Y. performed the data curation and investigation. S.X. wrote the original draft. G.G.W. reviewed and edited the manuscript. All authors have read and agreed to the published version of the manuscript.

**Funding:** This research was funded by the National Natural Science Foundation of China, grant numbers 41303064 and 41761063 and the Major Projects of Water Resources Department of Jiangxi Provincial grant, number KT201716.

**Acknowledgments:** The authors thank all the anonymous reviewers for their helpful remarks. Special thanks are also expressed to Hu Xianyi for the field observations, Liao Kaitao for the help in drawing the study area map, and Wu Jianping for the data processing and manuscript revision.

**Conflicts of Interest:** The authors have declared that no competing interests exist.

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
