# Peer review of "Effects of One-Year Simulated Nitrogen and Acid Deposition on Soil Respiration in a Subtropical Plantation in China"

_forests, doi:10.3390/f11020235_

Round 1
Reviewer 1 Report
General comments:
The manuscript needs a language revision.
In general, I find the manuscript interesting, I would, however, like to see better arguments for the actual treatments chosen. Also the discussion should be elaborated.
The random block design with three replicates seems to be an appropriate experimental design.
I miss information of the “natural” input of nitrogen and acid to the studied system (could be added in line 138). How are the amounts added to simulate nitrogen and acid deposition chosen? When choosing to add acid as a mixture of sulphuric and nitric acid, the effect of this alone might be more difficult to distinguish form the nitrogen addition. Actually, the main effect of both these forms of additions will mainly be to acidify the soil, since the soil is supposedly already saturated with nitrogen.
The statement in l. 239-240 that the inhibition effects of N was stronger than that of acid addition is really not very interesting, because it depends on the amounts added, which in this case might not be comparable.
It is quite strange that the combination of the two additions counteracts the effect of the N addition. I miss more discussion on this. Elaborate on the statement in l. 429-430.
Specific comments
l.29-30: Abbreviations should be explained.
l.35-37: Not quite clear what is cause and effect.
l.213: Normally you would expect the relationship between soil respiration and soil temperature to be exponential.
l.327-331: This paragraph is too complicated to read.
Reviewer 2 Report
See attached file
